# Low-Velocity Impact Response on Glass Fiber Reinforced 3D Integrated Woven Spacer Sandwich Composites

**DOI:** 10.3390/ma15062311

**Published:** 2022-03-21

**Authors:** Mahfuz Bin Rahman, Lvtao Zhu

**Affiliations:** 1College of Textile Science and Engineering (International Institute of Silk), Zhejiang Sci-Tech University, Hangzhou 310018, China; mahfuz.zstu@gmail.com; 2Shaoxing Keqiao Research Institute, Zhejiang Sci-Tech University, Shaoxing 312000, China

**Keywords:** low-velocity impact, glass fiber, epoxy resin, sandwich composites, damage mechanism

## Abstract

This study presents an experimental investigation on the low-velocity impact response of three-dimensional integrated woven spacer sandwich composites made of high-performance glass fiber reinforced fabric and epoxy resin. 3D integrated woven spacer sandwich composites with five different specifications were produced using a hand lay-up process and tested under low-velocity impact with energies of 5 J, 10 J, and 15 J. The results revealed that the core pile’s heights and diverse impact energies significantly affect the stiffness and energy absorption capacity. There is no significant influence of face sheet thickness on impact response. Moreover, the damage morphologies of 3D integrated woven spacer sandwich composites under different impact energies were analyzed by simple visualization of the specimen. Different damage and failure mechanisms were observed, including barely visible damage, visible damage, and clearly visible damage. Moreover, it was noticed that the damage of 3D integrated woven spacer sandwich composites samples only constraints to the impacted area and does not affect the integrity of the samples.

## 1. Introduction

Three-dimensional integrated woven spacer sandwich composites have been broadly utilized as modern materials for some applications attributable to their unrivaled physical and mechanical properties. Three-dimensional integrated woven spacer sandwich composites made of glass fiber with “8” shape and “1” shape structures in the core, having the advantages of being lightweight, energy absorbent, high strength, high stiffness, and cost-effective over the traditional sandwich structure composite materials. It is known as a new generation composite material [1]. It has been widely used for structural components in various areas, such as marine, aerospace, electrical, automotive, ballistic, building industries, medical areas, agricultural areas, and so on [2]. However, studying these composite materials’ failure and deformation processes is still necessary to discover the flaws and ways of improvement [3]. Nevertheless, the experimentation and advancement of composite materials and processes have become essential and famous since the 1940s [4]. In the last two decades, a substantial number of works on 3D integrated woven spacer sandwich composites in various aspects have been performed; some of those works can be found in references [5,6,7,8,9,10,11,12,13,14,15,16,17,18,19,20,21,22,23,24,25,26,27]. Despite that, a limited number of studies have focused on the impact response (drop weight impact) of 3D integrated woven spacer sandwich composites.

Li et al. [26,28] investigated the Charpy impact response at room and liquid nitrogen temperatures with six-core heights samples. Another study showed the static and dynamic mechanical conduct of 3D integrated woven spacer composites with thickened face sheets. Three-dimensional integrated woven spacer composites were prepared using the VARIM (vacuum-assisted resin infusion molding) process with polyester resins for both studies. It was found that under the Charpy impact test, the sample’s impact properties significantly improved at liquid nitrogen temperature compared to the room temperature. With the increase in core heights, the damage reduced, and the impact energy absorption capacity increased at liquid nitrogen and room temperatures. Moreover, it was found that under the static compression and dynamic low-velocity impact load, face sheet thickness significantly influences mechanical properties, damage, and failure mechanisms. Hosseini et al. [29] conducted finite element simulations and low-velocity impact tests under three energy levels with different specimens’ thickness of 3D woven hollow-core sandwich composites. They found that the perforation load and contact stiffness decreased with the sandwich panel thickness, whereas the energy absorption capacity increased. In addition, both FE and experimental results were in good agreement. Hosur et al. [30] presented the manufacturing process and the low-velocity impact properties of foam-filled 3-D integrated core sandwich composite laminates with hybrid face sheets. They provided a comprehensive review of impact performance applying additional face sheets developed by combining carbon/epoxy, S2-glass/epoxy, or a hybrid of the two. Impact test results revealed that additional face sheets remarkably improve the damage resistance and load-bearing capacity. Dubary et al. [31] investigated the impact response and damage tolerance aptitude of carbon and glass fiber reinforced woven-ply thermoplastic laminates at room temperature and near the glass transition temperature. From their results, it was obtained that temperature has a notable impact on the internal and external damages but has no significant influence on impact response with regard to maximum force and deflection. Andrew et al. [32] presented a critical review of fiber-reinforced polymer matrix composite material’s impact response concerning key parameters, e.g., geometry, materials, events, and environmental conditions. They showed that the fiber-reinforced polymer matrix composite materials parameters are vital aspects that affect the impact response. The desired performance of composite materials can be obtained by changing parameter aspects. Selecting the right parameters can better the impact resistance and performance of composites with competitive price reinforcements [33].

Impact resistance of composite materials is one of the most important and critical issues [34]. Therefore, in this study, the low-velocity impact (drop weight impact) response of glass fiber reinforced 3D integrated woven spacer sandwich composites have been examined experimentally. We aim to compare the impact response of different structured 3D integrated woven spacer sandwich composites under diverse impact energies. We mainly investigated structural parameter changes influence and the impact energies changes influence on drop weight impact performance. Furthermore, the damage morphology and failure mechanism are demonstrated.

## 2. Materials and Methods

For this study, composite sample materials used were the same as in previous publications, more details about sample materials can be found in the given reference [35]. Composite materials were supplied by Sinoma Science & Technology Co., Ltd. (Nanjing, China). Reinforcement materials used high-performance, alkali-free E-glass fiber, manufactured by Jushi Group Co., Ltd. (Jiaxing, China). Chemical compositions include silicon monoxide (SiO), dialuminum dioxide (Al_2_O_2),_ diboron dioxide (B_2_O_2)_, titanium dioxide (TiO_2)_, magnesium oxide (MgO), calcium oxide (CaO), ferric oxide (Fe_2_O_3_), sodium oxide + potassium oxide (Na_2_O + K_2_O) and fluorine (F). E-glass fiber diameter was 13 mm, and the linear density was 140 Tex. The weaving process produced the reinforcement fabric. The surface of E-glass fiber was coated by a silane-based infiltrant, which is ideal for the weaving process. This three-dimensional fabric structure can be divided into three layers: an upper layer, lower layer, and Z-directional fibers. As shown in Figure 1, the core piles Z-directional fibers warp surface exhibits an “8” shape, and the weft surface exhibits an “1” shape. 

Traditional hand lay-up technique used to prepare 3D integrated woven spacer sandwich composites. A schematic diagram of the hand lay-up technique with specifications is presented in Figure 2.

The epoxy resin is used as a matrix material for composite molding. The resin is named Huibo New Material 5078, supplied by Huibo New Material Technology Co., Ltd. (Shanghai, China). The weight ratio of the resin to the hardener was 4:1, and the weight ratio of resin to glass fiber was 1.1:1. The hand-lay-up process produced five different specifications of 3D integrated woven spacer sandwich composite boards. Three-dimensional integrated woven spacer reinforcement fabrics and sandwich composite samples specifications are given in Table 1 [35].

Drop weight impact is one of the most common low-velocity impact tests for composite materials [36]. Drop weight impact tests were conducted following ASTM D7136 [37]. The specimen’s size was modified to 100 mm × 100 mm based on the impact support fixture and clamps. For each group of samples, we prepared 6 specimens, and in total, 30 specimens for 5 groups of samples. The specimen for the drop weight impact test is shown in Figure 3.

Instron Dynatup 9250 impact test device was used for this experiment. This fully automated impact test device facilitates different impact energy ranges and a high-speed data acquisition system. 3D integrated woven spacer sandwich composites specimen was placed on the impact support fixture base and locked with four clamps to prevent moving during the impact test. The specimen was impacted at the center by a hemispherical striker tip, and the impact load, impact energy, deflection, and time data were recorded in the data acquisition system. The hemispherical intender diameter was 12.7 mm with a total weight of 7.225 kg. Figure 4a,b shows the Instron Dynatup 9250 impact test device and schematic diagram of clamping system and impactor geometrics.

## 3. Results and Discussion

This study focuses on the low-velocity impact (drop weight impact) response of three-dimensional integrated woven spacer sandwich composite materials under different impact energies. In total, five groups of samples with different core pile heights and face sheet thickness were tested under 5 J, 10 J, and 15 J impact energies. They correspond to the impact velocities of 1.1765 m/s, 1.6639 m/s, and 2.0378 m/s, which were chosen by controlling the impactor weight and height. In general, two types of impact damage can occur during a drop-weight impact test, one is clearly visible impact damage, and another one is barely visible impact damage [38]. Three different impact energies were chosen to test with (3 J) barely visible impact damage, (10 J) visible impact damage, and clearly visible impact damage. The impact energy can be calculated by the following equation [39]:(1)$$E_{i}=\frac{mv_{i}^{2}}{2}$$
where, $E_{i}$ = measured impact energy (J), *m* = mass of impactor (kg), and $v_{i}$ = impact velocity (m/s).

### 3.1. Force Deformation Curves

Force-deformation curves of 3D integrated woven spacer sandwich composites at all energies are plotted in Figure 5. It can be seen in all the groups of samples; the increment of impact energies causes the enhancement of maximum deformation. The initial deformation on composite materials signifies the change in stiffness, and the initial rising state in the force-deformation curve is termed stiffness [40]. The initial rising phase in the force-deformation curve relates to the load-bearing capacity of the composite sandwich panel specimens. The force-deformation curves show that the change in stiffness for each group of samples is different for diverse impact energy. The force-deformation curves at impact energy 5 J (Figure 5a) show a change in stiffness for the A group sample at a force level of 1.1 kN and deformation of 6 mm, for the B group sample at a force level of 1.6 kN and deformation of 6 mm, for the C group sample at a force level of 1.2 kN and deformation of 4.8 mm, for the D group sample at a force level of 1.5 kN and deformation of 5.2 mm and the E group sample at a force level of 1.08 kN and deformation of 5 mm. Although the top and bottom face sheets of the sample groups A, B, and C are the same with different core piles heights, like the top and bottom face sheets of the sample groups D and E are the same with different core piles height, the stiffness changes are clearly expressed under the same impact energy level. There is a trend to reduce the stiffness of the samples that can be observed with the increase in core piles height even though the increase in face sheet thickness.

Figure 5b,c also show a change in the stiffness and different structural degradation phenomena for all samples under the impact energies of 10 J and 15 J. The initial force and deformation of the sample groups A, B, C, D, and E under the impact energy 10 J (Figure 5b) are at a force level of 1.0 kN, 1.4 kN, 1.23 kN, 1.45 kN, and 1.12 kN and deformation of 5.0 mm, 4.8 mm, 4.9 mm, 6.0 mm, and 5.2 mm, respectively. Two different phenomena can be observed here when the core piles height is lower (sample group A and B); the structural degradation occurred earlier even though the force is lower compared to the force-deformation curves under impact energy 5 J. Moreover, the vice versa incident can be seen when the samples (sample group C, D, and E) core piles height gradually increase. The structural degradation takes longer even when the force is higher compared to the force-deformation curves under impact energy 5 J. In addition, the initial force and deformation of the sample groups A, B, C, D, and E under the impact energy 15 J (Figure 5c) is at a force level of 1.2 kN, 1.35 kN, 1.0 kN, 1.29 kN, and 1.3 kN and deformation of 7.0 mm, 5.0 mm, 5.03 mm, 5 mm, and 6.0 mm, respectively. Compare the force-deformation curves under impact energies 5 J, 10 J, and 15 J; the initial deformation increased with force increase. Moreover, it was discovered that the stiffness of the same sample changed under different impact energies. It is obvious from the outcomes that the impact response of 3D integrated woven spacer sandwich composites is influenced by core piles height. There was no significant influence of face sheet thickness on the stiffness under different impact energies.

In Figure 5 it can be seen after the initial degradation, the force-deformation curves fluctuate till the force and deformation reach the maximum. The maximum force and deformation of all groups of samples under the impact energies 5 J, 10 J, and 15 J can be seen in Table 2. 

In Figure 5a, the force-deformation curves of sample groups B and D slightly fluctuate, and the curves are in close form, which denotes no damage or barely visible damage on the top and bottom face sheet [41]. The sample may lose its stiffness in this case, and the impactor bounces back from the top face sheet. Therefore, the curves return from the maximum force and form a close type curve. In sample groups A, C, and E, the force-deformation curves fluctuation is more pronounced compared to the sample groups B and D. Moreover, sample groups A, C, and E show the descending section is a completely softening curve, which denotes the force value becomes zero at the end of the softening section. In this case, semi penetration or perforation occurs in the samples, and barely visible or visible damage occurs in the top and bottom sections. Figure 5a,b force-deformation curves show a noticeable fluctuation due to the various types of damage such as matrix cracks, fiber breakage, fiber interfacial debonding, etc. In Figure 5b, the A group sample shows an open type curve which means the complete penetration or perforation and clearly visible damage occurred in both the top and bottom sections. The rest of the four groups (sample group B, C, D, and E) shows the descending section is a completely softening curve, which denotes semi penetration or perforation takes place in the samples, and barely visible or visible damage occurred in the top and bottom section, unlike complete penetration or perforation. Figure 5c force-deformation curves represent an open type curve for all groups of samples, indicating complete penetration or perforation and clearly visible damage in both the top and bottom sections [42]. Moreover, from Table 2, it can be seen that the maximum deformation increased with the increase in impact energies. From the results, it is clear that under 5 J and 10 J impact energies, all groups of 3D integrated woven spacer sandwich composites undergo barely visible or visible structural degradation, and under 15 J impact energy the structural degradation is clearly visible.

### 3.2. Damage Morphology and Energy Absorption

Figure 6 shows the damage pictures of 3D integrated woven spacer sandwich composites under impact energies of 5 J, 10 J, and 15 J.

Figure 6 shows the barely visible, visible, and clearly visible damage under the diverse impact energies, as discussed in Section 3.1. Moreover, it can be seen from all the impact damaged samples images that the damage of 3D integrated woven spacer sandwich composite samples only constraints to the impacted area and does not affect the integrity of the samples. This demonstrates that such damage can be easily repairable to sustain the product’s service life. In Figure 6a under the 5 J impact energy, slight damage can be seen on the top face sheet of sample groups A, C, and E. There is slight damage also seen in the bottom face sheet of sample group A, but there is no visible damage on the bottom face sheet of sample groups C and E; this is due to the increment of core piles heights in sample groups C and E, which resist the bottom face sheet damage under 5 J impact energy. There is no damage on the top face sheet of sample group B, but a barely visible small fracture line can be seen on the bottom face sheet; this may occur due to sample bending during the impaction. There is a barely visible dent on the top face sheet in sample group D, but there is no visible damage on the bottom face sheet. In Figure 6b, under the impact energy of 10 J, visible impact damage can be seen on the top face sheet of all groups of samples. The bottom face sheet of sample group A cracks and delaminates outwards due to the low core piles height. The bottom face sheet of sample groups B, C, D, and E, has barely visible impact damage, either for sample bending during impaction or the densification of the core piles. In Figure 6c under the impact energy of 15 J, all groups of samples top face sheet completely perforated and bottom face sheet delaminated outwards. The deformation of all groups of samples, both top, and bottom face sheets, is clearly visible. It can be observed that the damage areas increased as the impact energy increased. It can be concluded that under the 5 J and 10 J impact energies, 3D integrated woven spacer sandwich composites can resist complete deformation, but under 15 J impact energy, it completely deforms. Figure 7 shows the energy absorption graph of three-dimensional integrated woven spacer sandwich composites before wholly structural degradation.

It can be seen from Figure 7 the energy absorption capacity of 3D integrated woven spacer sandwich composites gradually increases with the increase in impact energy, except the sample group A under the impact energy of 15 J. Moreover, it can be observed that the energy absorption capacity of the same group samples varies with the variation of impact energies. In Figure 7a,b, in all groups of samples under 5 J and 10 J impact energies, the energy absorption capacity is very close to impact energy. However, the energy absorption capacity of sample groups A and C are pretty distinct under the 15 J impact energy (Figure 7a). Three other groups’ sample energy absorption capacities are not even close to impact energy. Regardless, all group of samples undergoes maximum deformation under 15 J impact energy. Therefore, the maximum energy absorption limit of three-dimensional integrated woven spacer sandwich composites is 15 J under the drop weight impact test. In Figure 7c, the maximum energy absorption capacity is higher in sample group B, indicating that 3D integrated woven spacer sandwich composites can perform relatively well when the core piles height is at a certain level. Energy absorption capacity also increases with the increase in core piles heights. There is no significant influence of face sheet thickness on impact response. 

## 4. Conclusions

This study focused on the low-velocity impact response of E-glass reinforced three-dimensional integrated woven spacer sandwich composites. The individual impact behavior of the exact sample specifications under drop weight impact energies of 5 J, 10 J, and 15 J was investigated. As well, different sample specifications under different impact energies were investigated. The conducted tests showed that the increment of impact energies causes the enhancement of maximum deformation. The same sample’s stiffness parameter changed with variation of impact energies. Moreover, it was observed that the 3D integrated woven spacer sandwich composites stiffness decreased with the increase in core piles heights. Even the top face sheet thickness increased from 0.75 mm to 1.10 mm and bottom face sheet thickness increased from 0.48 mm to 0.80 mm with the increase in core piles heights; the stiffness degradation phenomena were similar. So, there is no significant influence of face sheet thickness on the initial impact response. In addition, the results showed that the energy absorption capacity increased with the increase in core piles heights and a certain level of core piles height 3D integrated woven spacer sandwich composites can perform relatively well under drop weight impact. Under simple visualization, it was seen that the damaged area expands with the increase in impact energies. The damage areas were more intense on low core piles heights samples than on high core piles heights samples. However, under the 5 J and 10 J impact energies, 3D integrated woven spacer sandwich composites can resist complete deformation even after different damages such as matrix cracks, fiber breakage, fiber interfacial debonding, etc. However, under 15 J impact energy, it completely deforms.

## Figures and Tables

**Figure 1 materials-15-02311-f001:**
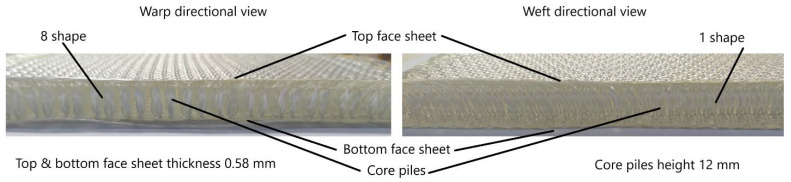
Three-dimensional integrated woven spacer fabric.

**Figure 2 materials-15-02311-f002:**
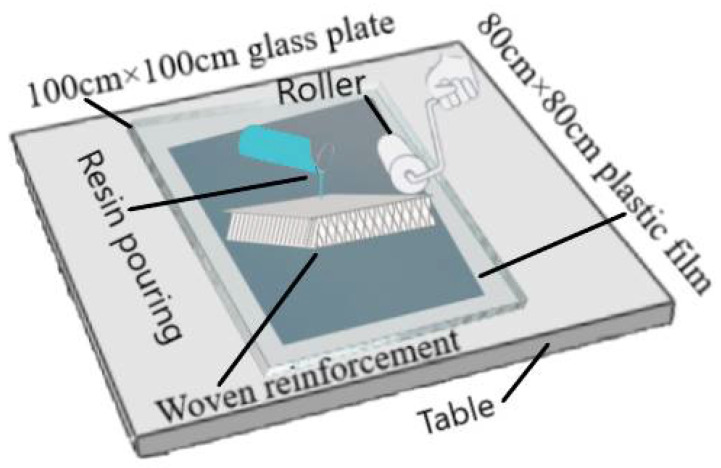
Schematic diagram of the hand lay-up technique.

**Figure 3 materials-15-02311-f003:**
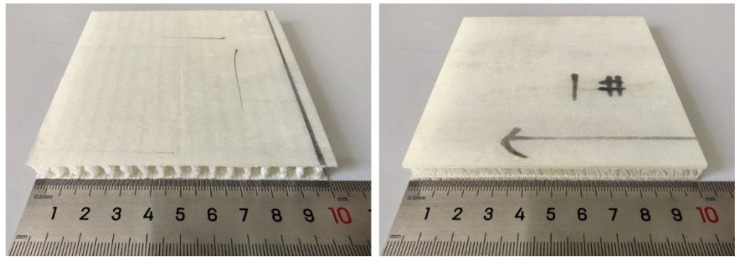
Drop weight impact test specimen.

**Figure 4 materials-15-02311-f004:**
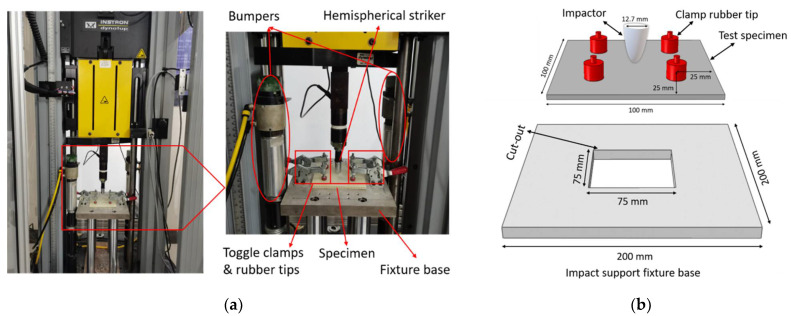
(**a**) Instron Dynatup 9250 impact test device. (**b**) Schematic diagram of clamping system and impactor geometrics.

**Figure 5 materials-15-02311-f005:**
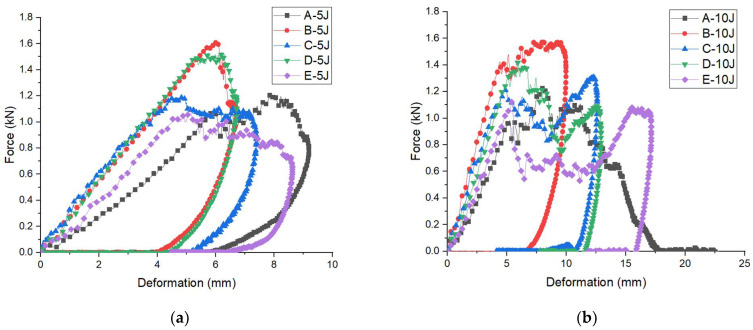
Force-deformation curves of five groups of 3D integrated woven spacer sandwich composites under different impact energy levels (**a**) 5 J, (**b**) 10 J, (**c**) 15 J.

**Figure 6 materials-15-02311-f006:**
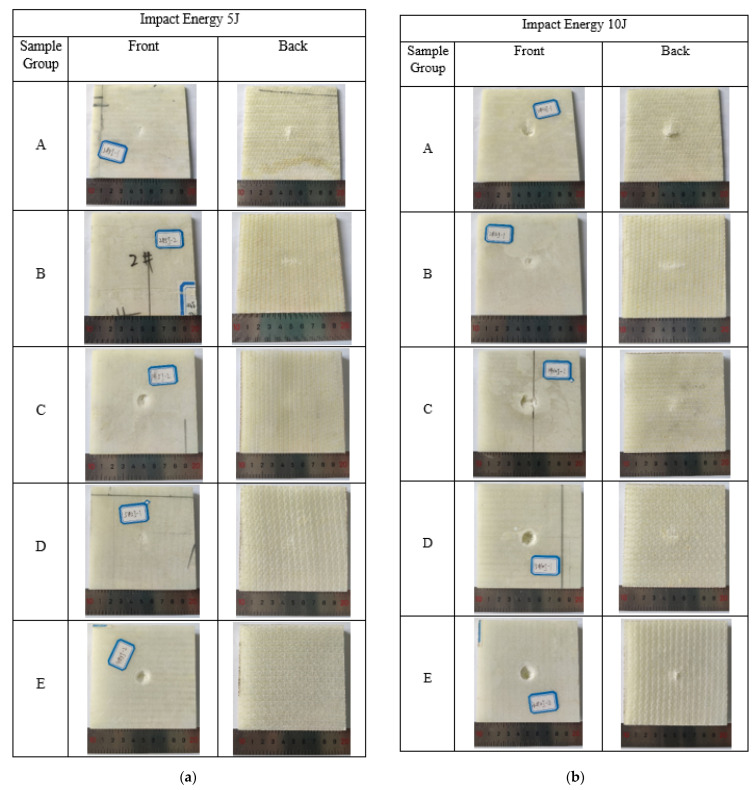
Impact damage samples under impact energies (**a**) 5 J, (**b**) 10 J, and (**c**) 15 J.

**Figure 7 materials-15-02311-f007:**
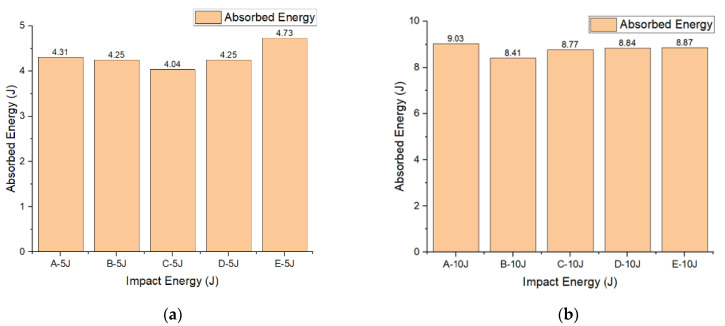
Energy absorption graph of 3D integrated woven spacer sandwich composites under impact energies (**a**) 5 J, (**b**) 10 J, and (**c**) 15 J.

**Table 1 materials-15-02311-t001:** Specifications of 3D integrated woven spacer reinforcement fabrics and sandwich composite samples.

Reinforcement Fabric Specifications	A	B	C	D	E
Core height (mm)	3.0	5.0	8.0	10.0	12.0
Top and bottom face sheet thickness (mm)	0.36	0.36	0.36	0.58	0.58
Fabric weight (g/m^2^)	720	810	920	1440	1440
Composite samples specifications	A	B	C	D	E
Total height (mm)	3.40	5.42	8.69	10.60	12.62
Core height (mm)	2.17	4.19	7.46	9.37	11.39
Top face sheet thickness (mm)	0.75	0.75	0.75	1.10	1.10
Bottom face sheet thickness (mm)	0.48	0.48	0.48	0.80	0.80
Warp piles density/inch	3	3	3	2	2
Weft piles density/inch	12	12	12	10	10
Composite weight (g/m^2^)	1510	1680	1900	2930	3060

**Table 2 materials-15-02311-t002:** Maximum force and deformation under the impact energies of 5 J, 10 J, and 15 J.

Sample Group	Impact Energy (5 J)		Impact Energy (10 J)		Impact Energy (15 J)	
	Peak Force (kN)	Max. Deformation (mm)	Peak Force (kN)	Max. Deformation (mm)	Peak Force (kN)	Max. Deformation (mm)
A	1.23	9.20	1.24	22.63	1.19	45.37
B	1.61	6.71	1.58	9.97	1.58	33.02
C	1.20	7.40	1.31	12.59	1.00	45.28
D	1.53	6.72	1.42	12.91	1.30	31.70
E	1.10	8.63	1.13	17.14	1.33	30.00

## Data Availability

Not applicable.

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
