# Peer review of "Low-Velocity Impact Response on Glass Fiber Reinforced 3D Integrated Woven Spacer Sandwich Composites"

_materials, 2022, doi:10.3390/ma15062311_

Round 1
Reviewer 1 Report
The authors have written well the manuscript. However, the manuscript needs major revision before it can be acceptable for publication. Specific comments are pointed out below:
- Reference bracket “( )” must be replace with “[ ]”.
- The introduction must be improved by adding other works related with state of art of newly impact test of composites materials. And the authors need more do review regarding impact test at low velocity of composite materials.
For examples:
https://link.springer.com/article/10.1007/s10443-022-10009-4 “Effects of Fiber Architectures on The Impact Resistance of Composite Laminates Under Low-Velocity Impact”
https://www.mdpi.com/2076-3417/12/4/1869 “Impact Properties of Novel Natural Fibre Metal Laminated Composite Materials”
https://doi.org/10.1016/j.paerosci.2021.100786 “On low-velocity impact behaviour of composite laminates: Damage investigation and influence of matrix and temperature”.
- It needs more illustration or add in Fig. 1 if possible. What is core height? Top & bottom face sheet thickness? Total height? Core height? Top face sheet thickness?
- 2 needs to be revised. Two digits number after comma is sufficient.
- 6 needs a scale bar. And wrong notation probably happens for Energy 5J and 10 J ?
- 7 needs to be improved or enlarged. Because the scale is too small. And why absorbed energy 10 J is higher than 15 J ?, Fig. 7(b) and (c) is probably reversed?
- It needs more discussion with other reference papers. “Figure 5 (c) force-deformation curves represent an open types curve for all 211 groups of samples, indicating complete penetration or perforation and clearly visible 212 damage in both the top and bottom sections”. Similar types of curves have been published elsewhere.
- Conclusion must be improved. What is the scientific finding based on authors’ study?

Reviewer 2 Report
Overall, this is an interesting paper that shows some empirical results about the damage sustained by woven composites due to impacts. I think overall the paper should be suitable for publication in this journal, but there are some issues with the quality of the writing. The authors should carefully consider going through the paper again to correct grammatical errors. Here are a few of the examples that I found, but there are others as well:
- Aimed to compare the impact... -> We aim to compare the impact
- Mainly investigated structural... -> We mainly investigated structural
- parameter changes influence drop weight -> how parameter changes influence drop weight
- materials used same as previous publication -> materials used were that same as in previous publication
- five different specifications 3D integrated -> five different specifications of 3D integrated
- impact test was conducted following -> impact tests were conducted following
- for each group of samples prepared 6 specimens -> for each group of samples we prepared 6 specimens
- even the force is lower -> even though the force is lower
- In figure 5 can be seen after -> In figure 5 it can be seen that after
- impact energy clearly visible structural degradation -> rewrite
- Demonstrates that such damage -> This demonstrates that such damage.
Here are a couple more issues to consider:
- Figure 2 is a little confusing. This schematic could be improved
- Table 2 better to have the whole table on one page so the headers are with the data
- How do you expect the results to change based on the method for producing the mats? For example, please consider the following paper: https://doi.org/10.3390/jcs4030124 Consider adding a reference to this paper and providing some insight on how the impact damage might be different for this system.
I hope these comments will be useful when revising the manuscript.
Reviewer 3 Report
Title: Low-velocity impact response on glass fiber reinforced 3D integrated woven spacer sandwich composites
Comments to authors
An interesting topic worthy of publishable in “Materials” after addressing the following minor comments:
- A well-written article. It should be proofread for some grammatical mistakes ad typos such as in the Abstract “There has no significant influence of face sheet thickness on impact response.” Should be written as “There is no significant influence of face sheet thickness on impact response.”…etc.
- Some numeric results should also be presented in the Abstract.
- The literature review section should be improved by adding some latest relevant references and discussions.
- Scope, Significance and Novelty should be highlighted in the last paragraph of the Introduction section.
- Line 108: “100mm*100mm” should be written as “100 mm x 100 mm”
- There should a space between the numerical values and the units. For example; “100mm*100mm” should be written as “100 mm x 100 mm”
- More discussions on the studied parameters are required and also it is recommended to add and discuss some other parameters related to mechanical performance and damaging response in the post-peak behavior of the glass fiber reinforced 3D integrated woven spacer sandwich composites.
- All the results should be supported by strong technical reasons and the relevant references.
- The conclusions section looks like a lab report. It should be presented by briefly writing the main findings of the present study.
Round 2
Reviewer 1 Report
Fig. 6 needs scale bar for all pictures.
And There are 4 Graphs for Fig. 7.
Best Wishes,
Author Response
Dear Reviewer,
On behalf of my coauthors, I would like to thank you for the suggestions and corrections to our manuscript. All the modifications have been marked by “Track changes”. Please see the below comments and the author's response.
Fig. 6 needs scale bar for all pictures.
Author’s Response: Thanks for your suggestion. We have added a scale bar with all the pictures.
And There are 4 Graphs for Fig. 7.
Author’s response: There are three graphs for figure 7. (a) for impact energy 5J, (b) for impact energy 10J, and (c) for impact energy 15J.
Reviewer 2 Report
The paper has been improved and can be accepted.
Author Response
Dear Reviewer,
On behalf of my coauthors, I would like to thank you for reviewing our manuscript and recommendations for acceptance.